# 5-Methyltetrahydrofolate Alleviates Memory Impairment in a Rat Model of Alzheimer’s Disease Induced by D-Galactose and Aluminum Chloride

**DOI:** 10.3390/ijerph192416426

**Published:** 2022-12-07

**Authors:** Zhengduo Zhang, Hong Wu, Shaojun Qi, Yanjin Tang, Chuan Qin, Rui Liu, Jiacheng Zhang, Yiyao Cao, Xibao Gao

**Affiliations:** 1Department of Physical and Chemical Inspection, School of Public Health, Cheeloo College of Medicine, Shandong University, Jinan 250012, China; 2Zhejiang Provincial Center for Disease Control and Prevention, Hangzhou 310051, China

**Keywords:** ad, toxicity assessment, human risk, rats, folic acid

## Abstract

The effects of 5-methyltetrahydrofolate (5-MTHF) on a rat model of Alzheimer’s disease (AD) induced by D-galactose (D-gal) and aluminum chloride (AlCl_3_) were investigated. Wistar rats were given an i.p. injection of 60 mg/kg D-gal and 10 mg/kg AlCl_3_ to induce AD and three doses of 1 mg/kg, 5 mg/kg or 10 mg/kg 5-MTHF by oral gavage. A positive control group was treated with 1 mg/kg donepezil by gavage. Morris water maze performance showed that 5 and 10 mg/kg 5-MTHF significantly decreased escape latency and increased the number of platform crossings and time spent in the target quadrant for AD rats. The administration of 10 mg/kg 5-MTHF decreased the brain content of amyloid β-protein 1-42 (Aβ_1-42_) and phosphorylated Tau protein (p-Tau) and decreased acetylcholinesterase and nitric oxide synthase activities. Superoxide dismutase activity, vascular endothelial growth factor level and glutamate concentration were increased, and malondialdehyde, endothelin-1, interleukin-6, tumor necrosis factor-alpha and nitric oxide decreased. The administration of 10 mg/kg 5-MTHF also increased the expression of disintegrin and metallopeptidase domain 10 mRNA and decreased the expression of β-site amyloid precursor protein cleavage enzyme 1 mRNA. In summary, 5-MTHF alleviates memory impairment in a D-gal- and AlCl_3_-exposed rat model of AD. The inhibition of Aβ_1-42_ and p-Tau release, reduced oxidative stress, the regulation of amyloid precursor protein processing and the release of excitatory amino acids and cytokines may be responsible.

## 1. Introduction

The prevalence of dementia is projected to double in Europe and triple globally by 2050, affecting 131 million people [1,2]. The most common form of dementia is Alzheimer’s disease (AD), an age-dependent neurodegenerative disease associated with metal exposure [3] and accompanied by impaired learning and memory, amyloid precursor protein (APP) and presenilin gene mutations, circadian rhythm disorders [4] and metabolic dysfunction [5]. Hypertension, hypercholesterolemia and dyslipidemia exacerbate AD, whereas walking and the Mediterranean diet reduce the risks [2,6]. AD continues to be an issue of concern due to the aging populations and elevated incidences in many countries [7]. Cholinesterase inhibitors and excitatory amino acid receptor antagonists such as donepezil and memantine have been approved for the treatment of AD [8]. Most other anti-amyloid β-proteins (Aβs), anti-Tau proteins and anti-inflammatory drugs have failed to produce positive results during clinical trials or are still in the early stages of clinical research. Therefore, the identification of drugs to treat AD has great practical and social significance.

Folic acid is supplied both from the diet and through intestinal microbial synthesis. It acts as a carrier of one-carbon units during metabolic reactions and protects against neural tube defects, megaloblastic anemia and cancer [9]. Supplementation with folic acid was shown to inhibit AD development and improve patients’ memory during a randomized controlled trial [10]. The possible explanations for this effect include the regulation of epigenetic modification (DNA methylation) [11], metabolism (lowering homocysteine concentration) [12], oxidative stress [13] and other signaling pathways. However, the dietary supplement form of folic acid cannot be converted into the biologically active form, 5-methyltetrahydrofolate (5-MTHF), unless methylenetetrahydrofolate reductase is present. About 26% of Chinese people lack this enzyme, reducing the benefits of folic acid supplements. 5-MTHF represents the predominant form of folic acid in human plasma and is the only form that is able to cross the blood–brain barrier [14]. In addition, 5-MTHF bioavailability is approximately seven times higher than that of folic acid [15]. Therefore, supplementation with 5-MTHF may be more beneficial than folic acid.

Little data regarding the effects of 5-MTHF on neurological conditions is available. The potential benefits of 5-MTHF for neuropathy in depressive patients [16] have been reported, but to the best of our knowledge, little is known about its impact on AD. The current study assesses the protective multi-target effects of 5-MTHF on a rat model of AD generated by exposure to D-galactose (D-gal) and aluminum chloride (AlCl_3_). Learning and memory functions are investigated. Levels of Aβ_1-42_, phosphorylated Tau protein (p-Tau) and amino acid neurotransmitters are measured and brain acetylcholinesterase (AChE), nitric oxide synthase (NOS) activities, serum oxidative stress and cytokine indices are evaluated. Hippocampal pathological sections and APP processing are also assessed. The aim is to expose potential therapies for AD patients with disorders of folate metabolism and to generate data for drug development.

## 2. Materials and Methods

### 2.1. Materials

D-gal (98% purity), AlCl_3_ (99% purity), 5-MTHF (≥95% purity) and donepezil (≥98% purity) were obtained from Macklin Biochemical Co., Ltd. (Shanghai, China). Aspartic acid (Asp), glycine (Gly), glutamate (Glu) and γ-aminobutyric acid (γ-GABA) reference standards (≥98% purity by HPLC) were obtained from Solarbio Science & Technology Co., Ltd. (Beijing, China). AChE, NOS, superoxide dismutase (SOD), nitric oxide (NO) and malondialdehyde (MDA) kits were obtained from the Nanjing Jiancheng Bioengineering Institute (Nanjing, China). Aβ_1-42_, p-Tau, endothelin-1 (ET-1), vascular endothelial growth factor (VEGF), interleukin-6 (IL-6) and tumor necrosis factor-alpha (TNF-α) ELISA kits were acquired from Jianglai Biological Technology Co., Ltd. (Shanghai, China). Primers for glyceraldehyde-3-phosphate dehydrogenase (GAPDH), disintegrin and metallopeptidase domain 10 (ADAM10) and β-site amyloid precursor protein cleavage enzyme 1 (BACE1) were purchased from General Biosystems Co., Ltd. (Anhui, China). Reverse transcription and PCR amplification kits were purchased from Accurate Biological Engineering Co., Ltd. (Hunan, China).

### 2.2. Animals and Treatments

A total of 72 specific pathogen-free (SPF) male Wistar rats (150–180 g) were acquired from Sibeifu Biotechnology Co., Ltd. (Beijing, China, animal certification number: SCXK (Jing) in 2019-0010 and housed at 22 ± 1 °C and 60 ± 5% humidity, with a 12 h light/12 h dark cycle and access to food and water ad libitum. Animals were allowed to adapt for one week in a clean, well-ventilated animal room before the formal experiment. All animal procedures were approved by the Animal Ethics Committee of Preventive Medicine of Shandong University (ID Number: LL20200802) and were carried out following China’s Animal Care and Use Guidelines. The rats were divided into six groups (n = 12) matched by body weight after adaptive feeding: Control, AD, AD + 1 mg/kg 5-MTHF, AD + 5 mg/kg 5-MTHF, AD + 10 mg/kg 5-MTHF and AD + donepezil groups. The model of AD was established by daily intraperitoneal (i.p.) injection of 60 mg/kg D-gal and 10 mg/kg AlCl_3_ for 6 weeks, as described previous [17,18]. AD + 1 mg/kg 5-MTHF, AD + 5 mg/kg 5-MTHF, AD + 10 mg/kg 5-MTHF and AD + donepezil groups were administered 60 mg/kg D-gal and 10 mg/kg AlCl_3_ intraperitoneally daily for 6 weeks, followed by the gavage of 1 mg/kg 5-MTHF, 5 mg/kg 5-MTHF, 10 mg/kg 5-MTHF and 1 mg/kg donepezil daily for 6 weeks, respectively [19]. Volume-matched saline was administrated to the control group by i.p. injection and gavage. Rats were weighed once a week and dose volumes adjusted according to body weight.

### 2.3. Morris Water Maze

The Morris water maze was utilized to evaluate learning and memory. It consists of a cylindrical pool with a diameter of 180 cm and a height of 80 cm, including a circular platform (diameter: 14 cm and height: 20 cm) in the second quadrant. A camera was installed above the pool to record swimming trajectories in real-time and transfer data to the operating system. The pool and surroundings were painted black, the four quadrants were indicated by space markers and light was blacked out with a curtain. The water temperature was maintained at 23–25 °C, and the platform was 1–2 cm below the water surface. Rats were put into the water from the four quadrants and the time was recorded to them finding the platform, up to a maximum of 60 s (escape latency). Animals were allowed to rest on the platform for 10 s if they found it within 60 s, but if not, the researcher would guide them to the platform to rest for 10 s and latency was recorded as 60 s. Daily latency was calculated as the mean of four trials. Rats were also put into the water from the fourth quadrant after the platform had been removed to swim for 60 s (probe task). The number of platform crossings and time spent in the target (second) quadrant were automatically recorded through the apparatus.

### 2.4. Tissue Processing

After the water maze tests were completed, all rats were made to fast for 24 h with water ad libitum. Two rats were chosen at random from each group for cardiac perfusion and brain tissue fixed with 4% paraformaldehyde for hematoxylin–eosin (HE) staining. The remaining rats were anesthetized with an i.p. injection of 10% chloral hydrate; blood was taken from the abdominal aorta and the serum separated by centrifugation at 3500 rpm for 10 min and stored at −80 °C. Brain tissue and hippocampus were snap-frozen and stored at −80 °C.

### 2.5. Determination of Aβ_1-42_, p-Tau, AChE and NOS

Brain tissue was added to saline pre-cooled at 4 °C in a weight:volume ratio of 1:9, 1 mm enzyme-free zirconia grinding beads was added for homogenization before centrifugation at 2500 rpm for 10 min and the supernatant used for measurements of Aβ_1-42_ and p-Tau concentrations and AChE and NOS activities. Assays were performed using kits in accordance with the manufacturers’ instructions.

### 2.6. Determination of Serum Antioxidant and Cytokine Indices

Then, 5% serum was used to determine SOD activity, 100% for MDA content and 100% for NO concentration; 20% serum was used to determine the VEGF, ET-1, IL-6 and TNF-α levels.

### 2.7. HE Staining of Hippocampus

Brain tissue was fixed with 4% paraformaldehyde and hippocampal tissue cut into 3 mm sections with a double-sided blade dehydrated with gradient ethanol (65% ethanol overnight, 75–85–95–100% ethanol for 2 h each) in an embedding box and cleaned in xylene. Tissue blocks were embedded, cut into 5 μm slices and dried on glass slides for 2 h in an oven at 60 °C. Paraffin sections were dewaxed (2 × 5 min washed with xylene), an ethanol gradient applied (100–95–85–75% for (2 min each), stained with HE for 5 min, differentiated in hydrochloric acid/ethanol for 30 s, stained with eosin for 2 min, dehydrated with 95% ethanol (1 min, two times) and absolute ethanol (1 min, two times), washed with xylene and sealed with neutral gum. Sections were washed with running water after hematoxylin, hydrochloric acid/ethanol and eosin application. Pathological changes to the hippocampus were observed with an inverted optical microscope (Olympus, Japan).

### 2.8. Determination of Brain Amino Acid Neurotransmitters

A 10% homogenate was made by adding 0.1 g brain tissue to 0.9 mL 50% acetonitrile and the supernatant used. Standard amino acid solutions of 5, 10, 50, 100, 200 and 400 μg/mL (300 μL) were measured, and the samples were mixed with 200 μL 0.5 mol/L sodium bicarbonate and 100 μL 0.5% 2,4-dinitrofluorobenzene (DNFB) and allowed to react in the dark for 55 min at 65 °C. Concentrations of Asp, Glu, Gly and γ-GABA were determined by HPLC (Shimadzu, Japan), with separation by an Intersil ODS-3 C18 column (4.6 × 250 mm, 5.0 μm) at 40 °C. The mobile phase was 0.05 mol/L sodium acetate (solvent A, PH = 6) and 50% acetonitrile (solvent B), with solvent B being increased from 12% to 45% in 0.1–4.0 min, increased from 45% to 60% in 4.0–20.0 min, decreased from 60% to 12% in 20.0–31.0 min and kept at 12% for 4.0 min, at a 1.0 mL/min flow rate. A photodiode array detector and a wavelength of 360 nm were used. The injection volume was 10 μL. Amino acid standard curve parameters are shown in Table 1.

### 2.9. RT-PCR Determination of the Hippocampal Expression of ADAM10 and BACE1 mRNA

Total RNA was extracted from 3 samples of hippocampal tissue randomly selected from each group, and the concentration and purity were assessed. Genomic DNA was removed and RNA reverse transcribed to cDNA. GAPDH (internal reference) cDNA was diluted 15-fold and ADAM10 and BACE1 cDNA 4-fold to ensure that the appropriate cycle threshold (Ct) was reached in the thermocycler. Primer sequences were as follows: GAPDH Forward: 5′ TCTCTGCTCCCCTCCCTGTTCT 3′; Reverse: 5′ ATCCGTTCACACCGACCTTC 3′; 95 bp; ADAM10 Forward: 5′ CAAAAACACCAGCGTGCCA 3′; Reverse: 5′ TCGTAGGTTGAATTGTCTTCCAT 3′; 93 bp; BACE1 Forward: 5′ GCAGTCAAGTCCATCAAGGC 3′; Reverse: 5′ GGCCGTAGGTATTGCTGAGGA 3′; 194 bp. PCR was performed as follows: preincubation at 95 °C for 30 s; PCR at 95 °C for 5 s, 60 °C for 30 s, 40 cycles; melting curve at 95 °C for 5 s, 60 °C for 1 min, increased to 95 °C; cooling at 40 °C for 15 s. Ct values were recorded, and the relative expression of target gene mRNA was calculated.

### 2.10. Statistical Analysis

Experimental results were tested for normality by the Shapiro–Wilk method and normally distributed data expressed as mean ± SEM. Escape latency was analyzed by repeated measures ANOVA and other normally distributed and equal variance data by one-way ANOVA. The Bonferroni test was utilized for pairwise comparison. Dunnett’s T3 test was utilized to verify the difference between groups of data with unequal variance. Significant differences were defined as *p* < 0.05 (two-tails). SPSS 26.0 software was used for all statistical analysis and GraphPad Prism 9.0.0 for the construction of figures.

## 3. Results

### 3.1. Effects of 5-MTHF on Memory Dysfunction in a Rat Model of AD

Escape latency gradually decreased with prolonged training time in the spatial navigation task for each group tested (F _time_ = 98.31, *p* < 0.001; F _group_ = 9.87, *p* < 0.001). A typical swimming trajectory on the fourth day of training for the six groups of rats is shown in Figure 1. Escape latency was more prolonged for AD rats than for healthy rats by days 2–4 (*p* < 0.05), indicating successful AD modeling. Time taken to find the target platform in AD rats dosed with 1, 5 or 10 mg/kg 5-MTHF or donepezil was lower than the untreated AD group on days 3–4 (*p* < 0.05, Figure 2a). Platform crossing number and time spent in the target quadrant were both reduced for AD rats compared with controls (*p* < 0.01), and all doses of 5-MTHF and donepezil increased these two indicators relative to untreated AD rats (*p* < 0.05, Figure 2b,c).

### 3.2. Effects of 5-MTHF on Brain Aβ_1-42_, p-Tau, AChE and NOS

Differences in brain Aβ_1-42_ and p-Tau are shown in Figure 3 (F = 16.17, *p* < 0.001; F = 12.85, *p* < 0.001). Aβ_1-42_ increased by 77.2% and p-Tau by 92.1% (*p* < 0.001) in AD rats relative to controls. Dosing AD rats with 1 mg/kg 5-MTHF reduced Aβ_1-42_ to 64.1%, with 5 mg/kg 5-MTHF to 64.8%, with 10 mg/kg 5-MTHF to 55.6% and with donepezil to 71.9% compared with untreated AD rats (*p* < 0.001). Similarly, dosing AD model rats with 5 mg/kg 5-MTHF reduced p-Tau to 66.6%, with 10 mg/kg 5-MTHF to 67.6% and with donepezil to 78.1% compared with untreated model rats (*p* < 0.05). AChE and NOS activities in the brains of AD rats were about 2.7 and 1.3 times those in controls, respectively (*p* < 0.001). Supplementation with 10 mg/kg 5-MTHF reduced AChE and NOS activities to 38.4% and 82.9% in the AD model rats (*p* < 0.05).

### 3.3. Effects of 5-MTHF on Serum Antioxidant and Cytokine Indices

Serum SOD activity decreased significantly in AD rats to about 77.9% of that of controls (*p* < 0.001). Treatment with 1 mg/kg 5-MTHF increased SOD by 23.5%, with 5 mg/kg 5-MTHF by 23.0%, with 10 mg/kg 5-MTHF by 30.1% and with donepezil by 36.0% compared with untreated AD rats (*p* < 0.05, Figure 4a). Serum MDA was about 1.7 times higher in AD rats than in controls (*p* < 0.001) and was decreased by 29.2% after treatment with 5 mg/kg 5-MTHF, by 30.9% with 10 mg/kg 5-MTHF and by 38.6% with donepezil (*p* < 0.01, Figure 4b). Levels of serum of ET-1 were 36.2% higher and of NO 35.8% higher in the AD model rats compared with controls (*p* < 0.001). ET-1 levels were reduced to 77.9% of those in the AD rats by 5 mg/kg 5-MTHF and to 72.1% by 10 mg/kg 5-MTHF (*p* < 0.01). NO levels were reduced as follows: 1 mg/kg 5-MTHF to 61.4% of values for AD rats, 5 mg/kg 5-MTHF to 56.1%, 10 mg/kg 5-MTHF to 62.2% and donepezil to 61.4% (*p* < 0.001). Serum TNF-α was 94.4% higher and IL-6 32.6% higher in the AD model than in controls (*p* < 0.001), but values were reduced as follows: 1 mg/kg 5-MTHF: TNF-α by 22.7%, IL-6 to 80.5%; 5 mg/kg 5-MTHF: TNF-α by 25.6%, IL-6 to 85.7%; 10 mg/kg 5-MTHF: TNF-α by 24.6%, IL-6 to 73.6%; donepezil: TNF-α by 22.9%, IL-6 to 81.3% (*p* < 0.05 for TNF-α, Figure 4e; *p* < 0.01 for IL-6, Figure 4f). Serum VEGF also decreased by 44.5% in AD rats compared with controls (*p* < 0.001) and was increased 1.7-fold by 10 mg/kg 5-MTHF and by 1.4-fold by donepezil treatment (*p* < 0.05, Figure 4g).

### 3.4. Effects of 5-MTHF on Hippocampal Neuronal Morphology

No abnormal hippocampal neurons were apparent in healthy rats (Figure 5), and the pyramidal cells of the CA1 region were neatly arranged and compact with clear boundaries. Pyramidal cell numbers decreased in AD rats and a loose, disordered arrangement with atrophied neurons could be seen. Treatment with 5-MTHF and donepezil improved the irregular pyramidal cell structure in the CA1 region.

### 3.5. Effects of 5-MTHF on Neurotransmitter Levels

Differences in Asp, Glu, Gly and γ-GABA are shown in Table 2 (F = 6.43, *p* < 0.001; F = 5.63, *p* < 0.001; F = 5.59, *p* < 0.001; F = 6.29, *p* < 0.001). Asp levels increased by 21.7% in AD rats relative to controls (*p* < 0.001) but were decreased by treatment with 5 mg/kg 5-MTHF to 82.3% and with donepezil to 86.1% (*p* < 0.05). Glu was 1.3-fold higher, Gly 1.4-fold higher and γ-GABA 1.7-fold higher in healthy rats than in the AD model rats (*p* < 0.05). Supplementation with 10 mg/kg 5-MTHF increased Glu by 28.0% (*p* = 0.002). HPLC separation chromatograms are shown in Figure 6.

### 3.6. Effects of 5-MTHF on Hippocampal ADAM10 and BACE1 mRNA

As shown in Figure 7, the expression of ADAM10 mRNA was lower in the hippocampus of AD model rats than in controls (*p* < 0.01) but was increased by 10 mg/kg 5-MTHF and donepezil treatment (*p* < 0.05). Expression of BACE1 mRNA was higher in the hippocampus of AD model rats than in controls (*p* = 0.007) and was decreased by treatment with 5 and 10 mg/kg 5-MTHF (*p* < 0.05).

## 4. Discussion

5-MTHF treatment was found to decrease memory dysfunction and restrict Aβ_1-42_ and p-Tau increases in a rat model of AD induced by D-gal and AlCl_3_ exposure, and the same treatment also ameliorated cholinergic damage and endothelial cell dysfunction, improved the numbers and structures of pyramidal cells in the hippocampal CA1 region and regulated oxidative stress and excitatory amino acid release.

Morris water maze testing illustrated the increased latency of AD rats, consistent with previous research [20], indicating successful AD modeling. These changes may be explained by the increased expression of Aβ in the brain in response to i.p. D-gal [21], which activates astrocyte proliferation (AS) [22] and causes neuronal damage [23] and memory impairment. Aluminum may inhibit cAMP-PKA-CREB signaling, leading to impaired synaptic plasticity and long-term potentiation damage [24]. The AD rats of the current study took longer to find the platform after 3–4 days than those treated with 1, 5 or 10 mg/kg 5-MTHF. The number of platform crossings and the time spent in the target quadrant were also lower in untreated AD rats compared with the 5-MTHF intervention group. In conclusion, 5-MTHF alleviated the memory dysfunction induced by D-gal and AlCl_3_ and delayed AD development.

Pathological changes involved in AD typically include senile plaques composed of accumulated Aβ deposits and neurofibrillary tangles containing p-Tau [25]. Aβ_1-42_ and p-Tau have been previously shown to increase in AD rats [26]. Increased Aβ_1-42_ may result from the decreased APP promoter methylation caused by AlCl_3_, which increases APP mRNA and promotes Aβ deposition [27]. In addition, Aβ-induced activation of glycogen synthase kinase-3β may cause the hyperphosphorylation of Tau protein [28]. Aβ_1-42_ and p-Tau were decreased by MTHF supplementation, and 5-MTHF may slow Aβ accumulation and p-Tau production by reducing APP levels and glycogen synthase kinase-3β activation, alleviating memory impairment.

Increased AChE activity was observed in the current cohort of AD rats, in agreement with previous findings [18], and the enzyme hydrolyzes synaptic acetylcholine, leading to cholinergic neurotransmission damage and cognitive decline. The administration of 10 mg/kg 5-MTHF decreased AChE activity. 5-MTHF may function as an acetylcholinesterase inhibitor by decreasing Aβ and promoting the generation of soluble products from APP [29]. NOS activity was elevated in AD rats, similar to previous results [30]. Aβ may stimulate nuclear factor-*κ*B (NF-*κ*B) [31] to enhance inducible NOS activity. NOS levels were decreased by 5 and 10 mg/kg 5-MTHF, perhaps due to suppression of the NF-*κ*B pathway.

Oxidative stress has been considered a risk factor for aging and various neurodegenerative diseases, including AD [32]. It may contribute to Aβ production and aggregation and increase p-Tau levels to aggravate AD. Decreased serum SOD activity and increased MDA levels were found in the AD rats of the current study, in agreement with previous work [33]. Increased Aβ may change mitochondrial dynamics and function by binding to the mitochondrial membrane to generate reactive oxygen species (ROS) [32], generating oxidative stress. Treatment with 5 and 10 mg/kg 5-MTHF raised serum SOD activity and lessened MDA levels compared with the model group. 5-MTHF may promote Aβ depolymerization, clearing ROS.

Endothelial cells are an essential part of the cardiovascular system, and they synthesize and release a variety of biologically active substances. However, abnormal cytokine secretion may lead to endothelial dysfunction, and this condition has been linked to AD [34]. Endothelial NO and ET-1 regulate vasodilation and contraction, and their levels are known to be higher in AD rats [30], consistent with the current results. Raised NO may result from elevated NOS activity, and ET-1 may be released in response to increased ROS levels [35]. TNF-α and IL-6 were found to be increased in AD rats [36] and VEGF to be decreased [37]. Aβ may activate AS, which stimulates p38 MAPK [38] to trigger NF-*κ*B activation [39] and aggravate the inflammatory response. Aβ also has anti-angiogenic activity, interacting with VEGF receptor 2 to block VEGF-mediated signaling [40]. 5-MTHF treatment reduced serum NO, ET-1, IL-6 and TNF-α and increased VEGF in AD rats, perhaps due to the reduced aggregation of Aβ and AS activation, which ameliorates endothelial and synaptic dysfunction [41].

The hippocampus of the brain’s limbic system is involved in learning, memory and spatial orientation [42], and the CA1 region is the most sensitive to damage. The hippocampal CA1 area has been shown to be atrophied in AD patients compared with normal individuals [43]. Changes to the numbers and arrangements of CA1 pyramidal cells and neuronal atrophy were observed in the AD rats of the present study. Neurotoxic Aβ accumulation may promote apoptosis and neuronal loss [44]. Adult hippocampal neurogenesis (AHN) may encourage hippocampal circuit plasticity and modulate hippocampal-dependent cognition [45]. AHN dysfunction is prevalent in AD patients and animal models [46]. The therapeutic effects of 5-MTHF during the current study may result from the inhibition of apoptosis and modulation of neurogenesis.

Brain amino acid neurotransmitters contribute to neuronal information transmission and cognitive activity [47]. Excitatory neurotransmitters, such as Asp and Glu, work together with inhibitory neurotransmitters, such as Gly and γ-GABA. Brain Asp levels were increased in AD rats, and Glu, Gly and γ-GABA were decreased [48]. Decreased glutamatergic and γ-GABAergic neuronal metabolism may result, and damage to the cholinergic system may be responsible for the increased release of excitatory Asp. 5-MTHF treatment decreased Asp and increased Glu, indicating the regulation of excitatory amino acids by 5-MTHF. Interestingly, 5-MTHF did not increase Gly and γ-GABA content, perhaps due to an insufficient time scale.

APP metabolism consists of the production of soluble products by the α-secretase ADAM10 and the production of Aβ by the β-secretase BACE1 [49]. Hippocampal ADAM10 and BACE1 mRNA expression levels are considered to be markers for AD progression. ADAM10 mRNA was reduced and BACE1 mRNA increased in the hippocampus of the AD model group, in agreement with previous studies [50]. ADAM10 is degraded by asparagine endopeptidase [51]. BACE1 expression may be upregulated by ROS production and lipid peroxidation following the oxidation of polyunsaturated fatty acids [52]. 5-MTHF supplementation raised ADAM10 mRNA and decreased BACE1 mRNA and may promote APP metabolism through the soluble product pathway.

A limitation of the current study concerns its multi-target approach to 5-MTHF, which covers a wide range of areas but does not include in-depth mechanistic analyses of AD pathogenesis. Further experiments are planned to scrutinize the mechanistic factors.

## 5. Conclusions

In conclusion, 5-MTHF alleviated memory impairment and restricted increases in Aβ_1-42_ and p-Tau in a rat model of AD induced by D-gal and AlCl_3_. SOD activity and VEGF levels were raised, and AChE and NOS activities and MDA, NO, ET-1, IL-6 and TNF-α levels were reduced by 5-MTHF treatment. The numbers and structures of pyramidal cells in the hippocampal CA1 region were restored and excitatory amino acid release and APP processing regulated.

## Figures and Tables

**Figure 1 ijerph-19-16426-f001:**
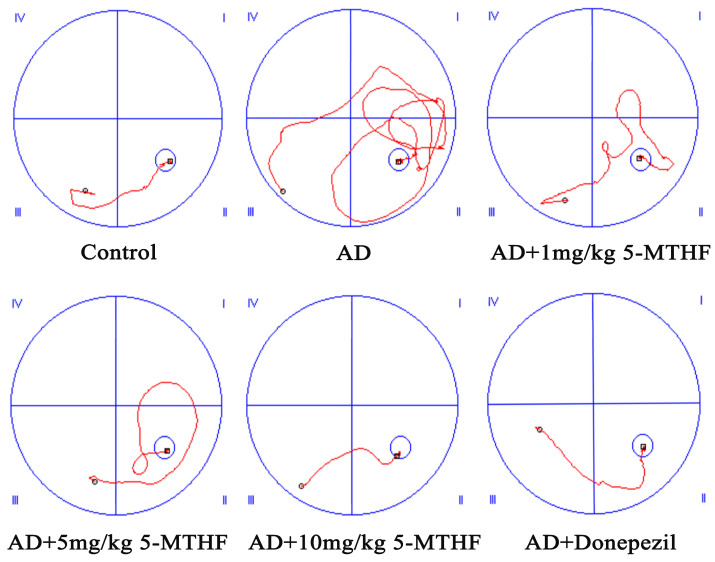
Typical swimming trajectory in the spatial navigation task for the six groups of rats (day 4). (I) First quadrant, (II) second quadrant, (III) third quadrant and (IV) fourth quadrant.

**Figure 2 ijerph-19-16426-f002:**
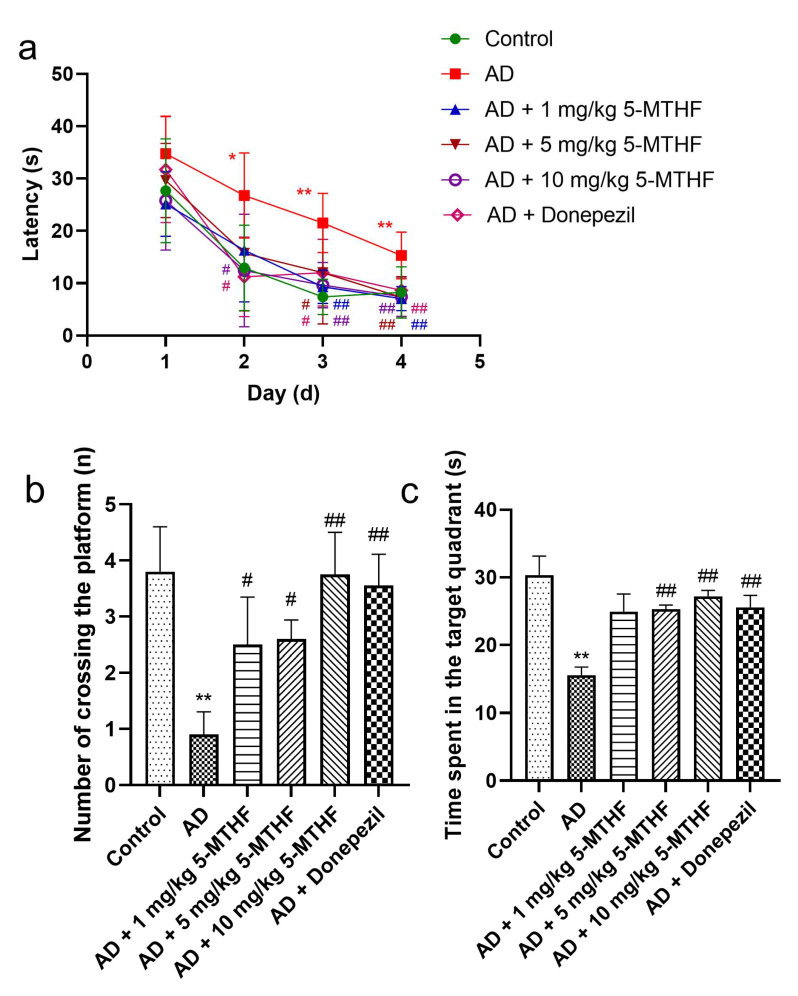
Effects of 5-MTHF on memory dysfunction of AD rats. (**a**) Latency in the spatial navigation task, (**b**) number of platform crossings and (**c**) time spent in the target quadrant. Data are shown as mean ± SEM. * *p* < 0.05, ** *p* < 0.01 compared with controls. ^#^ *p* < 0.05, ^##^ *p* < 0.01 compared with untreated AD rats.

**Figure 3 ijerph-19-16426-f003:**
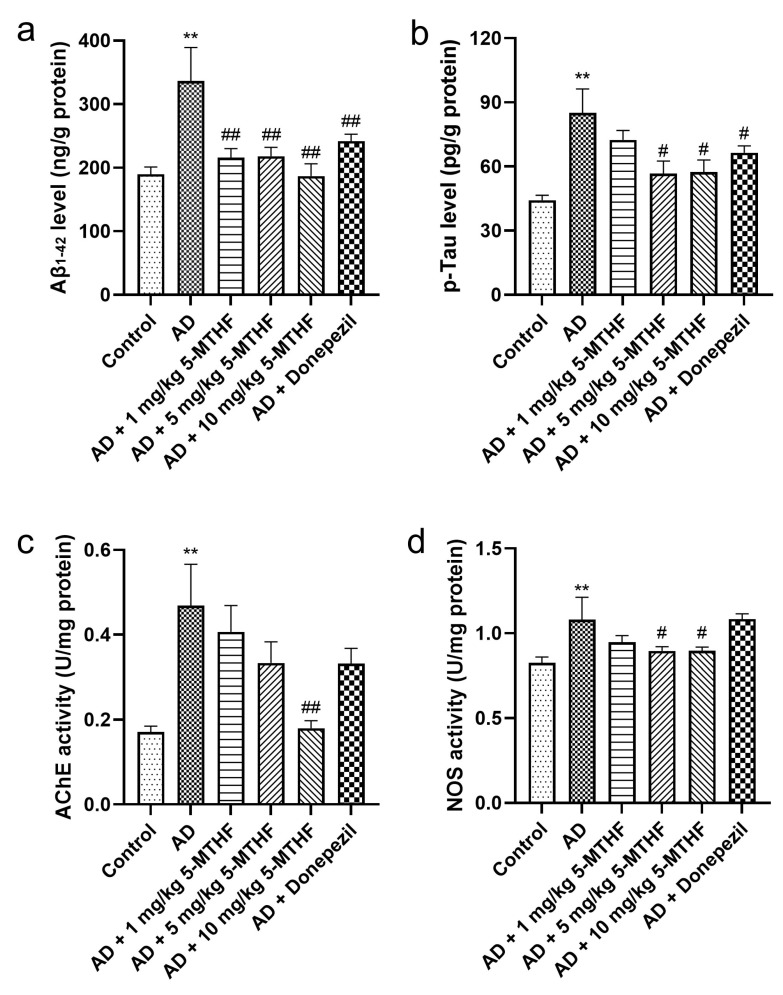
Aβ_1-42_ and p-Tau contents and AChE and NOS activities in brain tissue. (**a**) Aβ_1-42_, (**b**) p-Tau, (**c**) AChE and (**d**) NOS. Data are shown as mean ± SEM. ** *p* < 0.01 compared with controls. ^#^ *p* < 0.05, ^##^ *p* < 0.01 compared with the model group.

**Figure 4 ijerph-19-16426-f004:**
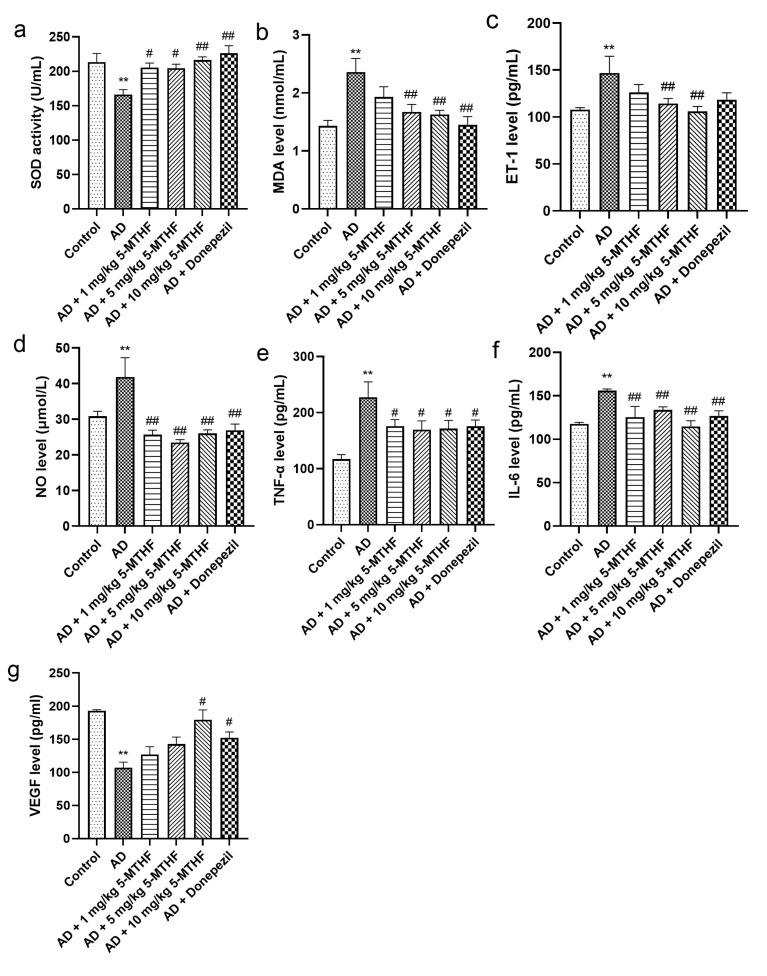
Serum antioxidant and cytokine indices. Antioxidants: (**a**) SOD activity and (**b**) MDA level. Cytokines: (**c**) ET-1, (**d**) NO, (**e**) TNF-α, (**f**) IL-6 and (**g**) VEGF. Data are shown as mean ± SEM. ** *p* < 0.01 compared with controls. ^#^ *p* < 0.05, ^##^ *p* < 0.01 compared with the model group.

**Figure 5 ijerph-19-16426-f005:**
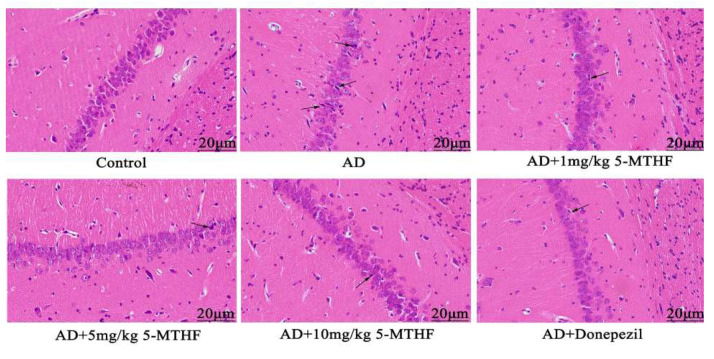
Typical HE staining images of the hippocampal CA1 region (magnification 400×). Atrophied neurons are indicated by arrows.

**Figure 6 ijerph-19-16426-f006:**
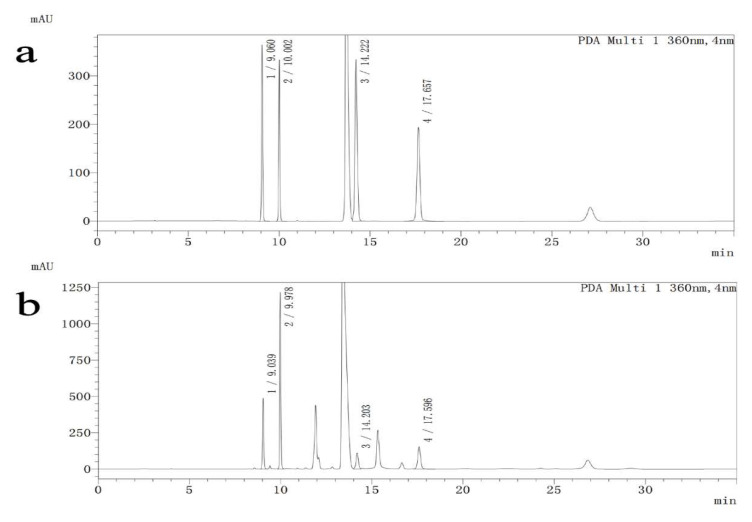
HPLC separation chromatograms for standard solutions and samples. (1) Asp, (2) Glu, (3) Gly, (4) γ-GABA. (**a**) Standard solutions, (**b**) Samples.

**Figure 7 ijerph-19-16426-f007:**
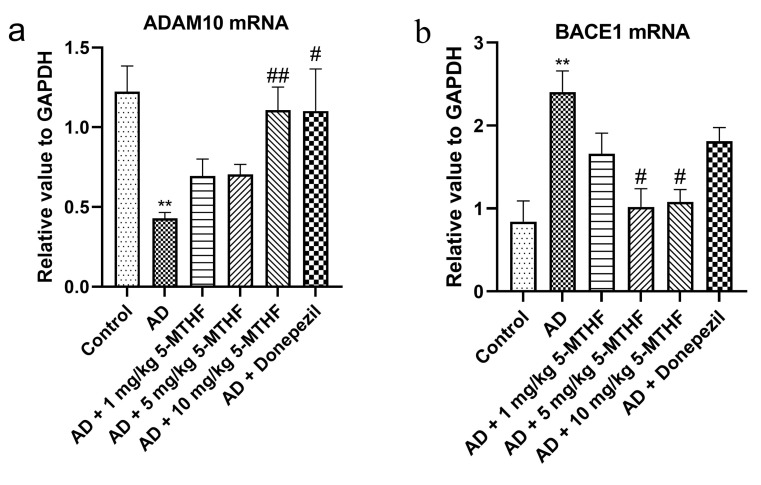
Effects of 5-MTHF on the expression of ADAM10 and BACE1 mRNA in the hippocampus of AD rats. (**a**) ADAM10 mRNA and (**b**) BACE1 mRNA. Data are shown as mean ± SEM. ** *p* < 0.01 compared with controls. ^#^ *p* < 0.05, ^##^ *p* < 0.01 compared with the model group.

**Table 1 ijerph-19-16426-t001:** Regression equations and correlation coefficients for Asp, Glu, Gly and γ-GABA.

Amino Acid	Regression Equation	Correlation Coefficient
Asp	A = 24,412C − 18,411	0.9992
Glu	A = 23,381C − 11,067	0.9991
Gly	A = 65,655C − 107,277	0.9998
γ-GABA	A = 46,910C − 91,513	0.9992

A: peak area; C: concentration.

**Table 2 ijerph-19-16426-t002:** Amino acid content of rat brain tissue (μg/mL, *n* = 10).

Group	Asp	Glu	Gly	γ-GABA
Control	71.27 ± 2.04	235.50 ± 6.86	21.62 ± 1.51	53.97 ± 2.73
AD	86.71 ± 1.30 **	188.18 ± 6.38 **	15.76 ± 0.43 *	31.80 ± 1.46 **
AD + 1 mg/kg 5-MTHF	78.08 ± 2.76	214.19 ± 6.76	16.67 ± 0.73	40.40 ± 2.64
AD + 5 mg/kg 5-MTHF	71.32 ± 2.45 ^##^	211.58 ± 8.70	16.00 ± 0.66	41.21 ± 2.59
AD + 10 mg/kg 5-MTHF	80.88 ± 3.28	240.80 ± 14.94 ^##^	17.91 ± 1.39	41.30 ± 5.54
AD + Donepezil	74.67 ± 2.90 ^#^	204.52 ± 8.13	16.78 ± 0.47	42.21 ± 2.77

Data are shown as mean ± SEM. * *p* < 0.05, ** *p* < 0.01 compared with controls. ^#^ *p* < 0.05, ^##^ *p* < 0.01 compared with the model group.

## Data Availability

The data presented in this study are available upon request from the corresponding author.

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
