# Peer review of "5-Methyltetrahydrofolate Alleviates Memory Impairment in a Rat Model of Alzheimer’s Disease Induced by D-Galactose and Aluminum Chloride"

_ijerph, 2022, doi:10.3390/ijerph192416426_

Round 1
Reviewer 1 Report
It's worth exploring AD interventions. Except for basic and necessary pathologic and toxicological examinations, readers would be more interested in how 5-MTHF modifies AD development and underlying mechanisms by combining in vivo and in vitro verification methods.
Reviewer 2 Report
1. The abstract should be more clear written.
For example : Instead “The Wistar rats were intraperitoneal injection of 60 mg/kg D-gal and 10 mg/kg AlCl3,while gavage 1 mg/kg, 5 mg/kg, 10 mg/kg 5-MTHF and 1 mg/kg donepezil.”
It can be written:
“The AD model in Wistar rats was induced by i.p. injection of 60 mg/kg D-gal and 10 mg/kg AlCl3. Treatment was done by gavage of 5-MTHF in three dose 1 mg/kg, 5 mg/kg, 10 mg/kg. Positive control was group of rats treated with 1 mg/kg donepezil by gavage.”
The Morris water maze shown that 5-MTHF significantly decreased the escape latency and increased the number of crossing the platform and the time spent in the target quadrant of AD rats. Did all doses have a significant effect?
2. The section 2.2. Animals and intervention in Material and Metods should be written more clearly and in more details.
For example: the model group was established for six weeks through intraperitoneal injection of 60 mg/kg D-gal and 10 mg/kg AlCl3. Is it a single dose or more doses,for how many days, it is not clear? 1 mg/kg 5-MTHF, 5 mg/kg 5-MTHF, 10 mg/kg 5-MTHF 94 and donepezil groups were gavaged 1 mg/kg 5-MTHF, 5 mg/kg 5-MTHF, 10 mg/kg 5- 95 MTHF and 1 mg/kg donepezil for 6 weeks after intraperitoneal injection of 60 mg/kg D- 96 gal and 10 mg/kg AlCl3 for 6 weeks, respectively. Single dose, or every day? Rewrite to be more clear.
3. Why don't you have a combined treatment group (5MTHF + donepezil)? Are you planning a combination therapy in the future?
4. 3.4. Effects of 5-MTHF on hippocampal neuronal morphology of AD rats
Figure 5 shows no apparent abnormal hippocampal neurons in healthy rats, and the pyramidal cells of the CA1 region are neatly arranged, compact, and have clear boundaries. The number of pyramidal cells of AD rats' CA1 area decreased, the arrangement was loose and disordered, and neurons atrophied. 5-MTHF and donepezil intervention proved the irregular structure of pyramidal cells in the CA1 region of AD rats. All doses, or just 10mg/kg of 5-MTHF? Is there difference between doses or not? Atrophied neurons should be point by arrows on the figure.
Reviewer 3 Report
The study is well designed, the results are well presented and written. I recommend the paper to be published. However, please carefully checks the line paragraph of reference No.31.
